# Information Thermodynamics for Time Series of Signal-Response Models

**DOI:** 10.3390/e21020177

**Published:** 2019-02-14

**Authors:** Andrea Auconi, Andrea Giansanti, Edda Klipp

**Affiliations:** 1Theoretische Biophysik, Humboldt-Universität zu Berlin, Invalidenstraße 42, D-10115 Berlin, Germany; 2Dipartimento di Fisica, Sapienza Università di Roma, 00185 Rome, Italy; 3INFN, Sezione di Roma 1, 00185 Rome, Italy

**Keywords:** irreversibility, fluctuation theorems, time series, transfer entropy, causal influence

## Abstract

The entropy production in stochastic dynamical systems is linked to the structure of their causal representation in terms of Bayesian networks. Such a connection was formalized for bipartite (or multipartite) systems with an integral fluctuation theorem in [Phys. Rev. Lett. 111, 180603 (2013)]. Here we introduce the information thermodynamics for time series, that are non-bipartite in general, and we show that the link between irreversibility and information can only result from an incomplete causal representation. In particular, we consider a backward transfer entropy lower bound to the conditional time series irreversibility that is induced by the absence of feedback in signal-response models. We study such a relation in a linear signal-response model providing analytical solutions, and in a nonlinear biological model of receptor-ligand systems where the time series irreversibility measures the signaling efficiency.

## 1. Introduction

The irreversibility of a process is the possibility to infer the existence of a time’s arrow looking at an ensemble of realizations of its dynamics [1,2,3]. This concept appears in the nonequilibrium thermodynamics quantification of dissipated work or entropy production [4,5,6], and it relates the probability of paths with their time-reversal conjugates [7].

Fluctuation theorems have been developed to describe the statistical properties of the entropy production and its relation to information-theoretic quantities in both Hamiltonian and Langevin dynamics [8,9,10]. Particular attention was given to measurement-feedback controlled models [11,12] inspired by the Maxwell’s demon [13], a gedanken-experiment in which mechanical work is extracted from thermodynamic systems using information. An ongoing experimental effort is put in the design and optimization of such information engines [14,15,16,17].

Theoretical studies clarified the role of fluctuations in feedback processes described by bipartite (or multipartite) stochastic dynamics, where fluctuation theorems set lower bounds on the entropy production of subsystems in terms of the Horowitz-Esposito information flow [18,19,20], or in terms of the transfer entropy [21,22] in the interaction between subsystems. Those inequalities form the second law of information thermodynamics [23], whose latest generalization was given in the form of geometrical projections into local reversible dynamics manifolds [24,25].

Time series can be obtained measuring continuous underlying dynamics at a finite frequency $\frac{1}{\tau }$, and this is the case of most real data. A measure of irreversibility for time series was defined in [26] as the Kullback–Leibler divergence [27] between the probability density of a time series realization and that of its time-reversal conjugate. Time series are non-bipartite in general, and this prevents an unambiguous identification and additive separation of the physical entropy production or heat exchanged with thermal baths [28]. Then the time series irreversibility does not generally converge to the physical bipartite (or multipartite) entropy production in the limit of high sampling frequency τ→0, except for special cases such as Langevin systems with constant diffusion coefficients, as we will discuss here.

The time series irreversibility measure defined in [26] depends on the statistical properties of the whole time series for non-Markovian processes. We define a measure of irreversibility that considers only the statistics of single transitions, and we call it *mapping irreversibility*. It recovers the irreversibility of whole time series only for Markovian systems. We study fluctuations of the mapping irreversibility introducing its stochastic counterpart. In the bivariate case of two interacting variables *x* and *y*, we study the conditional stochastic mapping irreversibility defined as the difference between that of the joint process and that of a single marginal subsystem [29].

We define signal-response models as continuous-time stationary stochastic processes characterized by the absence of feedback. In the bidimensional case, a signal-response model consists of a fluctuating signal *x* and a dynamic response *y*. In a recent work [30] we studied the information processing properties of linear (multidimensional) signal-response models. In that framework we defined a measure of causal influence to quantify how the macroscopic effects of asymmetric interactions are observed over time.

The backward transfer entropy is the standard transfer entropy [27] calculated in the ensemble of time-reversed trajectories. It was already shown to have a role in stochastic thermodynamics, in gambling theory, and in anti-causal linear regression models [21].

We derive an integral fluctuation theorem for time series of signal-response models that involves the backward transfer entropy. From this follows the II Law of Information thermodynamics for signal-response models, i.e., that the backward transfer entropy of the response *y* on the past of the signal *x* is a lower bound to the conditional mapping irreversibility. Please note that in general the conditional entropy production is zero lower bounded. Then our result shows that in time series it is the asymmetry of the interaction between subsystems *x* and *y*, namely the absence of feedback that links irreversibility with information.

For the basic linear signal-response model (BLRM) discussed in [30], in the limit of small observational time τ the backward transfer entropy converges to the causal influence. Also, in the BLRM, we find that the causal influence rate converges to the Horowitz-Esposito [18] information flow.

A key quantity here is the observational time τ, and the detection of the irreversibility of processes from real (finite) time series data is based on a fine-tuning of this parameter. We introduce such discussion with a biological model of receptor-ligand systems, where the entropy production measures the robustness of signaling.

The motivation of the present work is a future application of the stochastic thermodynamics framework to the analysis of time series data in biology and finance.

The paper is structured as follows. In Section 2 we provide a general introduction to stochastic thermodynamics introducing the irreversible entropy production. In Section 3 we state the setting and formalism we use for the treatment of bivariate time series, we define the stochastic (conditional) mapping irreversibility, we introduce the irreversibility density, and we review the general integral fluctuation theorem [31]. In Section 4 we discuss the integral fluctuation theorem for signal-response models involving the backward transfer entropy, and in Section 5.1 and Section 5.2 we show how it applies to the BLRM, and to a biological model of receptor-ligand systems. In the Discussion section we review the results and we motivate further our work. We provide an Appendix section where the analytical results for the mapping irreversibility, for the backward transfer entropy, and for the Horowitz-Esposito information flow in the BLRM are discussed.

## 2. Introduction to Continuous Information Thermodynamics

### 2.1. Entropy Production in Heat Baths

Let us consider an ensemble of trajectories generated by a Markovian (memoryless) continuous-time stochastic process composed of two interacting variables *x* and *y* subject to Brownian noise dW. The stochastic differential equations (SDEs) describing such kind of processes can be written in the Ito interpretation [32] as:(1)$$\begin{array}{c} \left\{\begin{array}{c} dx=g_{x}\left(x,y\right)dt+\sqrt{D_{x}\left(x,y\right)}\;\;dW_{x}\\ dy=g_{y}\left(x,y\right)dt+\sqrt{D_{y}\left(x,y\right)}\;\;dW_{y} \end{array}\right. \end{array}$$
where $D_{x}\left(x,y\right)$ and $D_{y}\left(x,y\right)$ are diffusion coefficients whose (x,y) dependence takes into account the case of multiplicative noise. Brownian motion is characterized by $\langle dW_{i}\left(t\right)dW_{j}\left(t^{\prime }\right)\rangle =\delta_{ij}\delta_{tt^{\prime }}dt$. The dynamics in (1) is bipartite, which means conditionally independent in updating: $p\left(x_{t+dt},y_{t+dt}|x_{t},y_{t}\right)=p\left(x_{t+dt}|x_{t},y_{t}\right)\cdot p\left(y_{t+dt}|x_{t},y_{t}\right)$.

The bipartite structure of (1) is fundamental in stochastic thermodynamics, because it allows the identification [28] and additive separation [18] of the heat exchanged with thermal baths in separate contact with *x* and *y* subsystems, $ds_{b}=ds_{b}^{x}+ds_{b}^{y}$. These are given by the detailed balance relation [5,10]:(2)$$ds_{b}^{x}=\ln\frac{p(x_{t+dt}|x_{t},y_{t})}{p(\tilde{x_{t+dt}}|\tilde{x_{t}},y_{t})},$$
where $\tilde{x_{t+dt}}$ is defined as the event of variable *x* assuming value $x_{t}$ at time t+dt, and similarly $\tilde{x_{t}}$ is the event of variable *x* assuming value $x_{t+dt}$ at time *t*. An analogous expression to (2) holds for subsystem *y*. Time-integrals of the updating probabilities $p(x_{t+dt}|x_{t},y_{t})$ and $p(\tilde{x_{t+dt}}|\tilde{x_{t}},y_{t})$ can be written in terms of the SDE (1) using Onsager-Machlup action functionals [19,33].

Stochastic thermodynamics quantities are defined in single realizations of the probabilistic dynamics, in relation to the ensemble distribution [31,34]. As an example, the stochastic joint entropy is $s_{xy}=-\ln p_{t}\left(x_{t},y_{t}\right)$, and its thermal (ensemble) average is the macroscopic entropy $S_{xy}={\langle s_{xy}\rangle }_{p_{t}\left(x_{t},y_{t}\right)}$. The explicit time dependence of $p_{t}$ describes the ensemble dynamics that in stationary processes is a relaxation to steady state. A SDE system such as (1) can be transformed into an equivalent partial differential equation in terms of probability currents [35], that is, the Fokker-Planck equation $\partial_{t}p\left(x,y,t\right)=-\partial_{x}J_{x}\left(x,y,t\right)-\partial_{y}J_{y}\left(x,y,t\right)$. Probability currents are related to average velocities with $J_{x}\left(x,y,t\right)\equiv p_{t}\left(x_{t}=x,y_{t}=y\right)\langle \dot{x_{t}}\rangle {|}_{x,y}$.

### 2.2. Feedbacks and Information

The stochastic entropy of subsystem *x unaware* of the other subsystem *y* is $s_{x}=-\ln p_{t}\left(x_{t}\right)$, and its time variation is $ds_{x}=\ln\frac{p_{t}\left(x_{t}\right)}{p_{t+dt}\left(x_{t+dt}\right)}$. The apparent entropy production of subsystem *x* with its heat bath is $ds_{x+b}=ds_{x}+ds_{b}^{x}$, and its thermal average $\langle ds_{x+b}\rangle$ can be negative due to the interaction with *y*, in apparent violation of the thermodynamics II Law. This is the case of Maxwell’s demon strategies [11,12], where information on *x* gained by the measuring device *y* is exploited to exert a feedback force and extract work from *x*, such as in the feedback cooling of a Brownian particle [19,36]. Integral fluctuation theorems [10,23] (IFTs) provide lower bounds on the subsystems’ macroscopic entropy production and extracted work in terms of information-theoretic measures [27]. The stochastic mutual information is defined as $I_{t}^{st}\equiv \ln\left(\frac{p_{t}\left(x_{t},y_{t}\right)}{p_{t}\left(x_{t}\right)p_{t}\left(y_{t}\right)}\right)$, where “st” stands for stochastic. Its time derivative can be separated into contributions corresponding to single trajectory movements and ensemble probability currents in the two directions, $d_{t}I_{t}^{st}=i_{t}^{x}+i_{t}^{y}$. The stochastic information flux in the *x* direction has the form $i_{t}^{x}=\dot{x_{t}}\partial_{x_{t}}\ln p_{t}\left(y_{t}|x_{t}\right)+\frac{\partial_{x}J_{x}\left(x,t\right)}{p_{t}\left(x_{t}\right)}-\frac{\partial_{x}J_{x}\left(x,y,t\right)}{p_{t}\left(x_{t},y_{t}\right)}$, where $J_{x}\left(x,t\right)=\int dy\;J_{x}\left(x,y,t\right)$. An analogous expression holds for $i_{t}^{y}$. The thermal average $I_{x\to y}\left(t\right)\equiv \langle i_{t}^{y}\rangle$ is the Horowitz-Esposito information flow [18,36]. At steady state it takes the form:(3)$$I_{x\to y}=\int \int dxdy\;J_{y}\left(x,y\right)\frac{\partial \ln p(x|y)}{\partial y}.$$
A recent formulation [19] upper bounds the average work extracted in feedback systems that in the steady-state bipartite framework is proportional to the *x* bath entropy change $\langle ds_{b}^{x}\rangle$, with the information flow (3) towards the *y* sensor. Such a result is recovered with a different formulation in terms of transfer entropies, and it is the Ito inequality [10,21,24] that reads:(4)$$\begin{array}{c} -\langle ds_{b}^{x}\rangle \leq T_{x\to y}\left(dt\right)-T_{x\to y}\left(-dt\right)=dtI_{x\to y}. \end{array}$$
where forward and backward stochastic transfer entropy [22] are respectively defined as $T_{x\to y}^{st}\left(dt\right)=\ln\left(\frac{p(y_{t+dt}|x_{t},y_{t})}{p(y_{t+dt}|y_{t})}\right)$, and $T_{x\to y}^{st}\left(-dt\right)=\ln\left(\frac{p(y_{t}|x_{t+dt},y_{t+dt})}{p(y_{t}|y_{t+dt})}\right)$.

### 2.3. Irreversible Entropy Production

The stochastic (total) *irreversible* entropy production [18,24,36] of the joint system and thermal baths is:(5)$$ds_{i}^{xy}=ds_{xy}+ds_{b}^{x}+ds_{b}^{y},$$
where $ds_{xy}=\ln\frac{p_{t}\left(x_{t},y_{t}\right)}{p_{t+dt}\left(x_{t+dt},y_{t+dt}\right)}$ is the joint system stochastic entropy change. If the ensemble is at steady state $p_{t+dt}\left(x_{t+dt},y_{t+dt}\right)=p_{t}\left(x_{t+dt},y_{t+dt}\right)=p\left(\tilde{x_{t}},\tilde{y_{t}}\right)$. If we further assume that diffusion coefficients in (1) are nonzero constants, and this is the case of Langevin systems [28] where these are proportional to the temperature, then the conditional probability $p(\tilde{x_{t+dt}}|\tilde{x_{t}},y_{t})$ is equivalent to $p\left(\tilde{x_{t+dt}}|\tilde{x_{t}},y\left(t\right)=y_{t+dt}\right)=p\left(\tilde{x_{t+dt}}|\tilde{x_{t}},\tilde{y_{t}}\right)$ under the time integral sign [10]. More precisely the term $\frac{1}{dt}\ln\left(\frac{p(\tilde{x_{t+dt}}|\tilde{x_{t}},y_{t})}{p(\tilde{x_{t+dt}}|\tilde{x_{t}},\tilde{y_{t}})}\right)$ almost surely vanishes. Then the irreversible entropy production (5) takes the form:(6)$$ds_{i}^{xy}=\ln\left(\frac{p(x_{t},y_{t},x_{t+dt},y_{t+dt})}{p(\tilde{x_{t}},\tilde{y_{t}},\tilde{x_{t+dt}},\tilde{y_{t+dt}})}\right).$$
Equation (6) shows the connection between entropy production and irreversibility of trajectories. The thermal average has the form of a Kullback–Leibler divergence [26,27] and satisfies $dS_{i}^{xy}\equiv \langle ds_{i}^{xy}\rangle \geq 0$. Using the Ito inequality [10,21,24] for both $ds_{b}^{x}$ and $ds_{b}^{y}$ does not lead to a positive lower bound in continuous (bipartite) stationary processes: $dS_{i}^{xy}\geq T_{x\to y}\left(-dt\right)-T_{x\to y}\left(dt\right)+T_{y\to x}\left(-dt\right)-T_{y\to x}\left(dt\right)=0$. Nevertheless, it is clear that the irreversible entropy production $dS_{i}^{xy}$ is strictly positive when the interaction between subsystems is nonconservative [9]. Our main interest is the stationary dissipation due to asymmetric interactions between subsystems, and how it is manifested in time series.

## 3. Bivariate Time Series Information Thermodynamics

### 3.1. Setting and Definition of Causal Representations

Let us assume that we can measure the state of the system (x,y) at a frequency $\frac{1}{\tau }$, thus obtaining time series. The finite observational time τ>0 makes the updating probability *not bipartite*: $p\left(x_{t+\tau },y_{t+\tau }|x_{t},y_{t}\right)=p\left(x_{t+\tau }|x_{t},y_{t},y_{t+\tau }\right)\cdot p\left(y_{t+\tau }|x_{t},y_{t}\right)=p\left(x_{t+\tau }|x_{t},y_{t}\right)\cdot p\left(y_{t+\tau }|x_{t},y_{t},x_{t+\tau }\right)$. Therefore, a clear identification of thermodynamics quantities in time series is not possible. Let us take the Markovian SDE system (1) as the underlying process, and let us further assume stationarity. Then the statistical properties of time series obtained from a time discretization can be represented in the form of Bayesian networks, where links correspond to the way in which the joint probability density $p(x_{t},y_{t},x_{t+\tau },y_{t+\tau })$ of states at the two instants *t* and t+τ is factorized. Still, there are multiple ways of such factorization. We say that a Bayesian network is a *causal representation* of the dynamics if conditional probabilities are expressed in a way that variables at time t+τ depend on variables at the same time instant or on variables at the previous time instant *t* (and not vice-versa), and that the dependence structure is done in order to minimize the total number of conditions on the probabilities. This corresponds to a minimization of the number of links in the Bayesian network describing the dynamics with observational time τ. Importantly, the causal representation is a way of factorizing the joint probability $p(x_{t},y_{t},x_{t+\tau },y_{t+\tau })$, and not a claim of causality between observables.

We define the combination $\zeta_{\tau }^{xy}$ as a pair of successive states of the joint system (x,y) separated by a time interval τ, $\zeta_{\tau }^{xy}\equiv \left(x\left(t\right)=x_{t},y\left(t\right)=y_{t},x\left(t+\tau \right)=x_{t+\tau },y\left(t+\tau \right)=y_{t+\tau }\right)\equiv f_{\tau }^{xy}\left(x_{t},y_{t},x_{t+\tau },y_{t+\tau }\right)\equiv \left(x_{t},y_{t},x_{t+\tau },y_{t+\tau }\right)$. We use the identity functional $f_{\tau }^{xy}\left(a,b,c,d\right)\equiv \left(x\left(t\right)=a,y\left(t\right)=b,x\left(t+\tau \right)=c,y\left(t+\tau \right)=d\right)$ for an unambiguous specification of the backward combination $\tilde{\zeta_{\tau }^{xy}}$. This is defined as the time-reversed conjugate of the combination $\zeta_{\tau }^{xy}$, meaning the inverted pair of the same two successive states, $\tilde{\zeta_{\tau }^{xy}}\equiv f_{\tau }^{xy}\left(x_{t+\tau },y_{t+\tau },x_{t},y_{t}\right)\equiv \left(\tilde{x_{t}},\tilde{y_{t}},\tilde{x_{t+\tau }},\tilde{y_{t+\tau }}\right)$. We defined backward variables of the type $\tilde{x_{t}}$ meaning $x\left(t\right)=x_{t+\tau }$, such correspondences being possible only when states at both times *t* and t+τ are given. The subsystems variables and backward variables are similarly defined as $\zeta_{\tau }^{x}=\left(x_{t},x_{t+\tau }\right)$, $\zeta_{\tau }^{y}=\left(y_{t},y_{t+\tau }\right)$, $\tilde{\zeta_{\tau }^{x}}=\left(\tilde{x_{t}},\tilde{x_{t+\tau }}\right)$, and $\tilde{\zeta_{\tau }^{y}}=\left(\tilde{y_{t}},\tilde{y_{t+\tau }}\right)$.

### 3.2. Definition of Mapping Irreversibility and the Standard Integral Fluctuation Theorem

A measure of coarse-grained entropy production for time series can be defined replacing dt with the nonzero observational time τ in the general expression (5) obtaining:(7)$$\begin{array}{c} \Delta s_{i}^{xy}=\Delta s_{xy}+\Delta s_{b}^{x}+\Delta s_{b}^{y}\equiv \\ \equiv \ln\frac{p(x_{t},y_{t})}{p(x_{t+\tau },y_{t+\tau })}+\ln\frac{p(x_{t+\tau }|x_{t},y_{t})}{p(\tilde{x_{t+\tau }}|\tilde{x_{t}},y_{t})}+\ln\frac{p(y_{t+\tau }|y_{t},x_{t})}{p(\tilde{y_{t+\tau }}|\tilde{y_{t}},x_{t})},\;\;\;\; \end{array}$$
where we assumed stationarity, $p_{t}=p$. By definition $\Delta s_{i}^{xy}$ converges to the physical entropy production in the limit τ→0, and it is a lower bound to it [37]. Importantly, such coarse-grained entropy production cannot have the form of an irreversibility measure such as (6) because $p\left(y_{t+\tau }|y_{t},x_{t},x_{t+\tau }\right)\neq p\left(y_{t+\tau }|y_{t},x_{t}\right)$. With “irreversibility form” we mean that its thermal average is a Kullback–Leibler divergence measuring the distinguishability between forward and time-reverse paths. Therefore, we decided to keep the widely accepted time series irreversibility definition given in [26], in its form for stationary Markovian systems. Anyway, we are interested in the time-reversal asymmetry of time series from even more general models or data where no identification of thermodynamic quantities is required.

For the study of fluctuations, we define the stochastic mapping irreversibility with observational time τ of the joint system (x,y) as:(8)$$\begin{array}{c} \varphi_{\tau }^{xy}=\ln\left(\frac{p(\zeta_{\tau }^{xy})}{p(\tilde{\zeta_{\tau }^{xy}})}\right). \end{array}$$
The thermal average is called mapping irreversibility, $\Phi_{\tau }^{xy}\equiv {\langle \varphi_{\tau }^{xy}\rangle }_{p(\zeta_{\tau }^{xy})}$, and it describes the statistical properties of a single transition over an interval τ. However, since the underlying dynamics (1) is Markovian, it describes the irreversibility of arbitrary long time series.

Let us note that $\varphi_{\tau }^{xy}$ does not generally converge to the total physical entropy production (5) in the limit τ→0 because of the non-bipartite structure of $p(x_{t+\tau },y_{t+\tau }|x_{t},y_{t})$ for any τ>0. It does converge anyway in most physical situations where bipartite underlying dynamics such as (1) has constant and strictly positive diffusion coefficients, and this is the case of Langevin systems. This is because the Brownian increments $dW_{x}$ and $dW_{y}$ are dominating the dynamics for small intervals τ, then the estimate of $g_{y}\left(x_{t+t^{\prime }},y_{t+t^{\prime }}\right)$ (with $0\leq t^{\prime }\leq \tau$) based on $\left(x_{t},y_{t}\right)$ is improved with the knowledge of $x_{t+\tau }$ just by a term of order $\partial_{x}g_{y}\left(x,y\right)\cdot W_{x}\left(\tau \right)∼\sqrt{\tau }$, where we assumed a smooth $g_{y}\left(x,y\right)$. Therefore in the limit τ→0 it is almost surely $p\left(y_{t+\tau }|x_{t},y_{t},x_{t+\tau }\right)\to p\left(y_{t+\tau }|x_{t},y_{t}\right)$ and $\varphi_{\tau }^{xy}\to \Delta s_{i}^{xy}\to ds_{i}^{xy}$, see Appendix D.

The stochastic mapping irreversibility satisfies the standard integral fluctuation theorem [31], i.e.,:(9)$$\begin{array}{c} {\langle e^{-\varphi_{\tau }^{xy}}\rangle }_{p(\zeta_{\tau }^{xy})}=\int_{\Omega }d\zeta_{\tau }^{xy}p\left(\tilde{\zeta_{\tau }^{xy}}\right)=1, \end{array}$$
where $d\zeta_{\tau }^{xy}=dx_{t}dy_{t}dx_{t+\tau }dy_{t+\tau }$, $\tilde{dx_{t}}=dx_{t+\tau }$, and Ω is the whole space of the combination $\zeta_{\tau }^{xy}$. From the convexity of the exponential function it follows that the mapping irreversibility $\Phi_{\tau }^{xy}$ is non-negative. This is the standard thermodynamics II Law inequality for the joint system (x,y) time series:(10)$$\begin{array}{c} \Phi_{\tau }^{xy}={\langle \varphi_{\tau }^{xy}\rangle }_{p(\zeta_{\tau }^{xy})}\geq 0. \end{array}$$
Similarly, we define the stochastic mapping irreversibility for the two subsystems as $\varphi_{\tau }^{x}\equiv \ln\left(\frac{p(\zeta_{\tau }^{x})}{p(\tilde{\zeta_{\tau }^{x}})}\right)$ and $\varphi_{\tau }^{y}\equiv \ln\left(\frac{p(\zeta_{\tau }^{y})}{p(\tilde{\zeta_{\tau }^{y}})}\right)$, these being called the marginals [29]. Their ensemble averages are respectively denoted $\Phi_{\tau }^{x}\geq 0$ and $\Phi_{\tau }^{y}\geq 0$, and they also satisfy the standard II Law.

Although bivariate time series derived from the joint process (1) are Markovian, the one-dimensional subsystems time series are generally not. This is because subsystems trajectories are a coarse-grained representation of the full dynamics, and to reproduce the statistical properties of those trajectories a non-Markovian dynamics has to be assumed. Therefore $\Phi_{\tau }^{x}$ and $\Phi_{\tau }^{y}$ are generally different from the irreversibility calculated on a whole time series. The mapping irreversibility $\Phi_{\tau }^{x}$ describes the statistical properties of the whole time series only if it is Markovian. This is surely the case if *x* is not influenced by *y* in (1), $\partial_{y}g_{x}\left(x,y\right)=\partial_{y}D_{x}\left(x,y\right)=0$, and motivated our study of signal-response models [30].

We define the conditional mapping irreversibility of *y* given *x* as the difference between the mapping irreversibility of the joint system (x,y) and the mapping irreversibility of system *x* alone:(11)$$\begin{array}{c} \Phi_{\tau }^{y|x}\equiv \Phi_{\tau }^{xy}-\Phi_{\tau }^{x}={\langle \ln\left(\frac{p(y_{t},y_{t+\tau }|x_{t},x_{t+\tau })}{p(\tilde{y_{t}},\tilde{y_{t+\tau }}|\tilde{x_{t}},\tilde{x_{t+\tau }})}\right)\rangle }_{p(\zeta_{\tau }^{xy})}. \end{array}$$
This can be considered as a time series generalization to the conditional entropy production introduced in [29]. Also, the conditional mapping irreversibility satisfies the standard integral fluctuation theorem and the corresponding II Law-like inequality $\Phi_{\tau }^{y|x}={\langle \varphi_{\tau }^{y|x}\rangle }_{p(\zeta_{\tau }^{xy})}\geq 0$.

In the general case where the evolution of each variable is influenced by the other variable (Equation (1)), we have a complete causal representation resulting from the dynamics (Figure 1), meaning that all edges are present in the Bayesian network. Please note that the horizontal arrows are non-directed because of the factorization degeneracy, meaning that the causal representation is given by multiple network topologies. In the complete case (Figure 1) we were not able to provide a more accurate characterization of the conditional mapping irreversibility $\Phi_{\tau }^{y|x}$ than the one given by the standard fluctuation theorem and the corresponding II Law-like inequality, $\Phi_{\tau }^{y|x}\geq 0$.

Let us recall that the inequalities for continuous bipartite systems [10,18,19,21,24] apply to the apparent entropy production $\Delta s_{x+b}=\Delta s_{x}+\Delta s_{b}^{x}$, and do not influence the total irreversible entropy production $\Delta s_{i}^{xy}$. Similarly, those results do not influence the mapping irreversibility $\Phi_{\tau }^{xy}$ and the conditional mapping irreversibility $\Phi_{\tau }^{y|x}$, for which in general only the standard zero lower bound can be provided.

We aim to relate the irreversibility of time series (8) to the (discrete) information flow between subsystems variables over time. We argue that fluctuation theorems linking the irreversibility of time series with information arise because of missing edges in the causal representation of the dynamics in terms of Bayesian networks. In the bivariate case there is only one class of time series generated from continuous underlying dynamics for which integral fluctuation theorems involving information measures can be written, and it corresponds to dynamics without feedback: the signal-response models.

### 3.3. Ito Inequality for Time Series

The Ito fluctuation theorem [10,21,24] for bipartite non-Markovian dynamics can be extended to Markovian non-bipartite time series if we modify the subsystems apparent entropy production $\Delta s_{y+b}=\Delta s_{y}+\Delta s_{b}^{y}$ into the explicitly non-bipartite form $\eta_{\tau }^{y|x}$:(12)$$\begin{array}{c} \eta_{\tau }^{y|x}=\ln\left(\frac{p(y_{t+\tau }|y_{t},x_{t},x_{t+\tau })}{p(\tilde{y_{t+\tau }}|\tilde{y_{t}},\tilde{x_{t}},\tilde{x_{t+\tau })}}\right)+\ln\left(\frac{p(y_{t})}{p(y_{t+\tau })}\right). \end{array}$$
Then the Ito fluctuation theorem for time series is written:(13)$$\begin{array}{c} {\langle e^{-\eta_{\tau }^{y|x}+T_{y\to x}^{st}\left(-\tau \right)-T_{y\to x}^{st}\left(\tau \right)+I_{xy}^{st}\left(t+\tau \right)-I_{xy}^{st}\left(t\right)}\rangle }_{p(\zeta_{\tau }^{xy})}=1. \end{array}$$
In stationary processes the mutual information is time invariant, $I_{xy}\left(t\right)=I_{xy}\left(t+\tau \right)$, and (13) implies the II Law-like inequality:(14)$$\begin{array}{c} \langle \eta_{\tau }^{y|x}\rangle \geq T_{y\to x}\left(-\tau \right)-T_{y\to x}\left(\tau \right). \end{array}$$
Similar to the apparent entropy production $\Delta s_{y}+\Delta s_{b}^{y}$ for bipartite systems, also $\langle \eta_{\tau }^{y|x}\rangle$ is not ensured to be positive if system *x* acts like a Maxwell’s demon.

The definition (12) does not have a clear physical meaning in time series, apart from its convergence to the apparent entropy production for τ→0, again for a continuous underlying dynamics with constant nonzero diffusion coefficients. In addition, $\langle \eta_{\tau }^{y|x}\rangle$ does not have the form of a Kullback–Leibler divergence, and is then not considered a measure of irreversibility. Therefore, in the following we will be interested instead in the conditional mapping irreversibility $\Phi_{\tau }^{y|x}$ (Equation (11)). Importantly, there is no general connection between $\Phi_{\tau }^{y|x}$ and information measures, and such connection will result instead from the topology of the causal representation in signal-response models.

### 3.4. The Mapping Irreversibility Density

Let us use an equivalent representation of the mapping irreversibility in terms of backward probabilities [38] defined as $p_{B}\left(\zeta_{\tau }^{xy}\right)=p\left(x\left(t\right)=x_{t},y\left(t\right)=y_{t},x\left(t-\tau \right)=x_{t+\tau },y\left(t-\tau \right)=y_{t+\tau }\right)$. For stationary processes it holds $p_{B}\left(\zeta_{\tau }^{xy}\right)=p\left(\tilde{\zeta_{\tau }^{xy}}\right)$ and $\varphi_{\tau }^{xy}=\ln\left(\frac{p(\zeta_{\tau }^{xy})}{p_{B}\left(\zeta_{\tau }^{xy}\right)}\right)$. We introduce here the mapping irreversibility density (with observational time τ) for stationary processes as:(15)$$\begin{array}{c} \psi \left(x_{t},y_{t}\right)=\int_{-\infty }^{\infty }\int_{-\infty }^{\infty }dx_{t+\tau }dy_{t+\tau }p\left(\zeta_{\tau }^{xy}\right)\varphi_{\tau }^{xy}=\int_{-\infty }^{\infty }\int_{-\infty }^{\infty }dx_{t+\tau }dy_{t+\tau }p\left(\zeta_{\tau }^{xy}\right)\ln\left(\frac{p(\zeta_{\tau }^{xy})}{p_{B}\left(\zeta_{\tau }^{xy}\right)}\right)=\\ =p\left(x_{t},y_{t}\right)\int_{-\infty }^{\infty }\int_{-\infty }^{\infty }dx_{t+\tau }dy_{t+\tau }p\left(x_{t+\tau },y_{t+\tau }|x_{t},y_{t}\right)\ln\left(\frac{p(x\left(t+\tau \right)=x_{t+\tau },y\left(t+\tau \right)=y_{t+\tau }|x\left(t\right)=x_{t},y\left(t\right)=y_{t})}{p(x\left(t-\tau \right)=x_{t+\tau },y\left(t-\tau \right)=y_{t+\tau }|x\left(t\right)=x_{t},y\left(t\right)=y_{t})}\right).\;\;\;\; \end{array}$$
The mapping irreversibility density $\psi (x_{t},y_{t})$ tells us which situations $\left(x_{t},y_{t}\right)$ contribute more to the time series irreversibility of the macroscopic process. $\psi (x_{t},y_{t})$ is proportional to the distance (precisely to the Kullback–Leibler divergence [27]) of the distribution of future states $p(x_{t+\tau },y_{t+\tau }|x_{t},y_{t})$ to the distribution of past states $p(x_{t-\tau },y_{t-\tau }|x_{t},y_{t})$ of the same condition $\left(x_{t},y_{t}\right)$.

## 4. The Fluctuation Theorem for Time Series of Signal-Response Models

If the system (x,y) is such that the variable *y* does not influence the dynamics of the variable *x*, then we are dealing with signal-response models (Figure 2). The stochastic differential equation for signal-response models is written in the Ito representation [32] as:(16)$$\begin{array}{c} \left\{\begin{array}{c} dx=g_{x}\left(x\right)dt+\sqrt{D_{x}\left(x\right)}\;\;dW_{x}\\ dy=g_{y}\left(x,y\right)dt+\sqrt{D_{y}\left(x,y\right)}\;\;dW_{y} \end{array}\right. \end{array}$$
The absence of feedback is written in $\frac{\partial g_{x}}{\partial y}=\frac{\partial D_{x}}{\partial y}=0$. As a consequence, the conditional probability satisfies $p\left(y_{t}|x_{t},x_{t+\tau }\right)=p\left(y_{t}|x_{t}\right)$, and the corresponding causal representation is incomplete, see the Bayesian network in Figure 2. In other words, signal-response models are specified by the property that $x_{t+\tau }$ is conditionally independent on $y_{t}$ given $x_{t}$.

For signal-response models we can provide a lower bound on the entropy production that is more informative than Equation (10), and that involves the backward transfer entropy $T_{y\to x}\left(-\tau \right)$. The backward transfer entropy [21] is a measure of discrete information flow towards the past, and is here defined as the standard transfer entropy for the ensemble of time-reversed combinations $\zeta_{\tau }^{xy}$. The stochastic counterpart as a function of $\zeta_{\tau }^{xy}\backslash y_{t}$ is defined as:(17)$$\begin{array}{c} T_{y\to x}^{st}\left(-\tau \right)=\ln\left(\frac{p(x_{t}|y_{t+\tau },x_{t+\tau })}{p(x_{t}|x_{t+\tau })}\right), \end{array}$$
where st stands for stochastic.

Then by definition $T_{y\to x}\left(-\tau \right)={\langle T_{y\to x}^{st}\left(-\tau \right)\rangle }_{p(\zeta_{\tau }^{xy}\backslash y_{t})}$. We keep the same symbol $T_{y\to x}$ as the standard transfer entropy because in stationary processes the backward transfer entropy is the standard transfer entropy (calculated on forward trajectories) for negative shifts −τ.

The fluctuation theorem for time series of signal-response models is written:(18)$$\begin{array}{c} {\langle e^{-\varphi_{\tau }^{xy}+\varphi_{\tau }^{x}+T_{y\to x}^{st}\left(-\tau \right)}\rangle }_{p(\zeta_{\tau }^{xy})}=\\ =\int_{\Omega }d\zeta_{\tau }^{xy}p\left(\tilde{y_{t+\tau }}|\tilde{x_{t}},\tilde{y_{t}},\tilde{x_{t+\tau }}\right)p\left(x_{t},x_{t+\tau },y_{t+\tau }\right)=1,\;\;\;\; \end{array}$$
where we used the signal-response property of no feedback $p\left(\tilde{y_{t}}|\tilde{x_{t}},\tilde{x_{t+\tau }}\right)=p\left(\tilde{y_{t}}|\tilde{x_{t}}\right)$, the stationarity property $p\left(\tilde{y_{t}}|\tilde{x_{t}}\right)=p\left(y_{t+\tau }|x_{t+\tau }\right)$, the correspondence $dy_{t}=\tilde{dy_{t+\tau }}$, and the normalization property $\int_{-\infty }^{\infty }d\tilde{y_{t+\tau }}p\left(\tilde{y_{t+\tau }}|\tilde{x_{t}},\tilde{y_{t}},\tilde{x_{t+\tau }}\right)=1$.

From the convexity of the exponential it follows the II Law of Information thermodynamics for time series of signal-response models:(19)$$\begin{array}{c} \Phi_{\tau }^{y|x}\equiv \Phi_{\tau }^{xy}-\Phi_{\tau }^{x}\geq T_{y\to x}\left(-\tau \right), \end{array}$$
and this is the central relation we wish to study in the Applications section hereafter.

In the limit of τ→0 and constant nonzero diffusion coefficients, $\Phi_{\tau }^{y|x}$ converges to the heat exchanged with the thermal bath attached to subsystem *y*, $\langle ds_{m}^{y}\rangle$. Therefore, the inequality (19) converges to the Ito inequality (14) in its form for Markovian signal-response bipartite systems [21,24] as expected. Indeed, in signal-response models $\Phi_{\tau }^{y|x}=\langle \eta_{\tau }^{y|x}\rangle$, $T_{y\to x}\left(\tau \right)=0$, and (14) transforms into (19). Therefore (19) can be regarded as an extension to (non-bipartite) time series of the Ito inequality [10,21,24]. This shows that in time series it is the asymmetry of the interaction between subsystems *x* and *y*, namely the absence of feedback, that links irreversibility with information.

Please note that $\Phi_{\tau }^{x}$ is equivalent to the original time series irreversibility [26] because the *x* time series is Markovian in the absence of feedback.

In causal representations of correlated stationary processes, the factorization of $p(x_{t},y_{t})$ is unnecessary, and only the structure of the transition probability $p(x_{t+\tau },y_{t+\tau }|x_{t},y_{t})$ has to be specified. Please note that the direction of the horizontal $x_{t}$-$y_{t}$ arrow is never specified (see Figure 1 and Figure 2). In the complete (symmetric) case with feedback we also do not specify the direction of the horizontal $x_{t+\tau }$-$y_{t+\tau }$ arrow because of the full degeneracy (see Figure 1). The importance of the causal representation is seen in signal-response models (Figure 2) because we could have decomposed the transition probability as well into the non-causal decomposition $p\left(x_{t+\tau },y_{t+\tau }|x_{t},y_{t}\right)=p\left(y_{t+\tau }|x_{t},y_{t}\right)\cdot p\left(x_{t+\tau }|x_{t},y_{t},y_{t+\tau }\right)$, but this does not lead to the fluctuation theorem (18).

## 5. Applications

### 5.1. The Basic Linear Response Model

We study the II Law for signal-response models (Equation (19)) in the BLRM, whose information processing properties are already discussed in [30]. The BLRM is composed of a fluctuating signal *x* described by the Ornstein-Uhlenbeck process [39,40], and a dynamic linear response *y* to this signal:(20)$$\begin{array}{c} \left\{\begin{array}{c} dx=-\frac{x}{t_{rel}}dt+\sqrt{D}\;\;dW\\ \frac{dy}{dt}=\alpha x-\beta y \end{array}\right. \end{array}$$
The response *y* is considered in the limit of weak coupling with the thermal bath $D_{y}\to 0$, while the signal is attached to the source of noise, $D_{x}=D>0$.

This model allows analytical representations for the mapping irreversibility $\Phi_{\tau }^{xy}$ (calculated in Appendix A) and the backward transfer entropy $T_{y\to x}\left(-\tau \right)$ (calculated in Appendix B). We find that once the observational time τ is specified, $\Phi_{\tau }^{xy}$ and $T_{y\to x}\left(-\tau \right)$ are both functions of just the two parameters $t_{rel}$ and β, which describe respectively the time scale of the fluctuations of the signal and the time scale of the response to a deterministic input. In addition, if we choose to rescale the time units by $t_{rel}$ to compare fluctuations of different timescales, we find that irreversibility measures are function of just the product $\beta t_{rel}$ that is then the only free parameter in the model.

Since the signal is a time-symmetric (reversible) process, $\Phi_{\tau }^{x}=0$, the backward transfer entropy $T_{y\to x}\left(-\tau \right)$ is the lower bound on the total entropy production $\Phi_{\tau }^{xy}$ in the BLRM.

The plot in Figure 3 shows the mapping irreversibility $\Phi_{\tau }^{xy}$ and the backward transfer entropy $T_{y\to x}\left(-\tau \right)$ as a function of the observational time τ. In the limit of small τ, the entropy production diverges because of the deterministic nature of the response dynamics (the standard deviation on the determination of the velocity $\frac{dy}{dt}$ due to instantaneous movements of the signal vanishes as $\alpha \sqrt{D}\sqrt{dt}\to 0$). The backward transfer entropy $T_{y\to x}\left(-\tau \right)$ instead vanishes for τ→0 because the Brownian motion has nonzero quadratic variation [32] and is the dominating term in the signal dynamics for small time intervals. In the limit of large observational time intervals τ→∞ the entropy production is asymptotically double the backward transfer entropy that is its lower bound given by the II Law for signal-response models (Equation (19)), $\frac{\Phi_{\tau }^{xy}}{T_{y\to x}\left(-\tau \right)}\to 2$. Interestingly, this limit of 2 is valid for any choice of the parameters in the BLRM.

Let us recall the definition of causal influence $C_{x\to y}\left(\tau \right)$ as a measure of information flow for time series [30]:(21)$$\begin{array}{c} C_{x\to y}\left(\tau \right)\equiv I\left(x\left(t\right),y\left(t+\tau \right)\right)-R\left(\tau \right)\equiv \\ \equiv I\left(x\left(t\right),y\left(t+\tau \right)\right)-\frac{1}{2}\ln\left(\frac{e^{2(I\left(x_{t},y_{t}\right)+I\left(y_{t+\tau },\left(x_{t},y_{t}\right)\right)}}{e^{2I(x_{t},y_{t})}+e^{2I(y_{t+\tau },\left(x_{t},y_{t}\right))}-1}\right). \end{array}$$
R(τ) is the redundancy measure quantifying that fraction of time-lagged mutual information $I(x_{t},y_{t+\tau })$ that the signal $x_{t}$ gives on the evolution $y_{t+\tau }$ of the response that is already contained in the knowledge of the current state $y_{t}$ of the response [42].

Interestingly, for small observational time τ→0, the causal influence of the signal on the evolution of the response converges to the backward transfer entropy of the response on the past of the signal $C_{x\to y}\left(\tau \right)\to T_{y\to x}\left(-\tau \right)$, and they both vanish with τβ. Also, the causal influence rate defined as $\lim_{\tau \to 0}\frac{C_{x\to y}\left(\tau \right)}{\tau }$ converges to the Horowitz-Esposito [18] information flow $I^{x\to y}$ (details in Appendix C).

For large observational time τ→∞ instead the causal influence converges to the standard (forward) transfer entropy $C_{x\to y}\left(\tau \right)\to T_{y\to x}\left(\tau \right)$. Also, in this limit τ→∞, the causal influence is an eighth of the entropy production $\frac{\Phi_{\tau }^{xy}}{C_{x\to y}\left(\tau \right)}\to 8$ for any choice of the parameters in the BLRM.

In general, the limit τ→∞ corresponds to recording the system state at a rate that is much slower compared to any stationary dynamics, so that only an exponentially small time-delayed information flow is observed. Similarly, time asymmetries in the dynamics become less visible and the irreversibility measures vanish.

Let us now consider the mapping irreversibility density $\psi (x_{t},y_{t})$ in the BLRM for small and large observational time τ. In Figure 4 we choose a τ smaller than the characteristic response time $\frac{1}{\beta }$ and smaller than the characteristic time of fluctuations $t_{rel}$. In the limit τ→0 the signal dynamics is dominated by noise and the entropy production is mainly given by movements of the response *y*. The two spots correspond to the points where the product of the density $p(x_{t},y_{t})$ times the absolute value of the instant velocity $\dot{y}$ is larger. For longer intervals $\tau ⪆\frac{1}{\beta }$ (that is the case of Figure 5) the history of the signal becomes relevant because it determined the present value of the response, therefore the irreversibility density is also distributed on those points of the diagonal (corresponding to roughly $\dot{y_{t}}=0$) where the absolute value of the signal $x_{t}$ is big enough. Also, consequently, in this regime the backward transfer entropy is a meaningful lower bound to the entropy production, that is Equation (19).

### 5.2. Receptor-Ligand Systems

The receptor-ligand interaction is the fundamental mechanism of molecular recognition in biology and is a recurring motif in signaling pathways [43,44]. The fraction of activated receptors is part of the cell’s representation of the outside world, it is the cell’s estimate on the concentration of ligands in the environment, upon which it bases its protein expression and response to external stress.

If we could experimentally keep the concentration of ligands fixed we would still get a fluctuating number of activated receptors due to the intrinsic stochasticity of the macroscopic description of chemical reactions. Recent studies allowed a theoretical understanding of the origins of the macroscopic “noise” (i.e., the output variance in the conditional probability distributions), and raised questions about the optimality of the input distributions in terms of information transmission [45,46,47,48].

Here we consider the dynamical aspects of information processing in receptor-ligand systems [49,50], where the response is integrated over time. If the perturbation of the receptor-ligand binding on the concentration of free ligands is negligible, that is in the limit of high ligand concentration, receptor-ligand systems can be modeled as nonlinear signal-response models [51]. We write our model of receptor-ligand systems in the Ito representation [32] as:(22)$$\begin{array}{c} \left\{\begin{array}{c} dx=-\left(x-1\right)dt+x\;dW_{x}\\ dy=k_{on}\left(1-y\right)\frac{x^{h}}{1+x^{h}}dt-k_{off}ydt+y\left(1-y\right)dW_{y}\;\;\; \end{array}\right. \end{array}$$
The fluctuations of the ligand concentration *x* are described by a mean-reverting geometric Brownian motion, with an average 〈x〉=1 in arbitrary units. The response, that is the fraction of activated receptors *y*, is driven by a Hill-type interaction with the signal with cooperativity coefficient *h*, and chemical bound/unbound rates $k_{on}$ and $k_{off}$. For simplicity, the dynamic range of the response is set to be coincident with the mean value of the ligand concentration that means choosing a Hill constant K=〈x〉=1. The form of the *y* noise is set by the biological constraint 0<y<1. For simplicity, we choose a cooperativity coefficient of h=2 that is the smallest order of sigmoidal functions.

The mutual information between the concentration of ligands and the fraction of activated receptors in a cell is a natural choice for quantifying its sensory properties [52]. Here we argue that in the case of signal-response models, the conditional entropy production is the relevant measure, because it quantifies how the dynamics of the signal produces irreversible transitions in the dynamics of the response, which is closely related to the concept of causation. Besides, our measure of causal influence [30] has yet not been generalized to the nonlinear case, while the entropy production has a consistent thermodynamical interpretation [31].

We simulated the receptor-ligand model of Equation (22), and we evaluated numerically the mapping irreversibility $\Phi_{\tau }^{xy}$ and the backward transfer entropy $T_{y\to x}\left(-\tau \right)$ using a multivariate Gaussian approximation for the conditional probabilities $p(x_{t+\tau },y_{t+\tau }|x_{t},y_{t})$ (details in Appendix E). The II Law for signal-response models sets $\Phi_{\tau }^{xy}\geq T_{y\to x}\left(-\tau \right)$ and proves to be a useful tool for receptor-ligand systems, as it is seen if Figure 6. Please note that the numerical estimation of the entropy production requires statistically many more samples compared to the backward transfer entropy: $\Phi_{\tau }^{xy}$ depends on $\zeta_{\tau }^{xy}$ (4 dimensions) while $T_{y\to x}\left(-\tau \right)$ depends on $\zeta_{\tau }^{xy}\backslash y_{t}$ (3 dimensions). In a real biological experimental setting, the sampling process is expensive, and the backward transfer entropy is therefore a useful lower bound for the entropy production, and an interesting characterization of the system to be used when the number of samples is not large enough.

The intrinsic noise of the response $y\left(1-y\right)dW_{y}$ is the dominant term in the response dynamics for small intervals τ, and $xdW_{x}$ is the dominant term for the signal. This makes both $\Phi_{\tau }^{xy}$ and $T_{y\to x}\left(-\tau \right)$ vanish in the limit τ→0. In the limit of large observational time τ, as it is also the case for the BLRM and in any stationary process, the entropy production for the corresponding time series $\Phi_{\tau }^{xy}$ and all the information measures are vanishing, because the memory of the system is damped exponentially over time by the relaxation parameter $k_{off}$ (β in the BLRM). Therefore, to better detect the irreversibility of a process one must choose an appropriate observational time τ. In the receptor-ligand model of Equation (22) with parameters $k_{on}=5$, $k_{off}=1$ and h=2 we see that the optimal observational time is around τ≈0.5 (see Figure 6). Here for "optimal" we mean the observational time that corresponds to the highest mapping irreversibility $\Phi_{\tau }^{xy}$. In general, one might be interested in inferring the irreversibility rate (that is $\frac{\Phi_{\tau }^{xy}}{\tau }$ in the limit τ→0) looking at time series data with finite sampling interval τ. In the receptor-ligand model of Figure 6 the irreversibility rate converges to 2.

## 6. Discussion

To put in perspective our work let us recall that the well-established integral fluctuation theorem for stochastic trajectories [34] leads to a total irreversible entropy production with zero lower bound, that is the standard II Law of thermodynamics. Modern inequalities such as the II Law of Information thermodynamics [21,23,24] describe how the information continuously shared between subsystems can lead to an "apparent" negative entropy production in (one of) the subsystems. Nevertheless, these do not bring to any difference in the total joint irreversible entropy production whose lower bound is still zero.

Our aim here was to characterize cases in which more informative lower bounds on the total irreversible entropy production can be provided. Ito-Sagawa [10] already showed that for Bayesian controlled systems (where a parameter can be varied to perform work) a general fluctuation theorem for the subsystems and the relative lower bound on entropy production is linked to the topology of the Bayesian network representation associated with the stochastic dynamics of the system. This connection seems to be even stronger in the case of the total joint (uncontrolled) system that is the object of our study. We show in the bidimensional case of a pair of signal-response variables how a missing arrow in the Bayesian network describing the dynamics leads to a fluctuation theorem.

The detailed fluctuation theorem linking work dissipation and the irreversibility of trajectories in nonequilibrium transformations [5,8] holds in mechanical systems attached to heat reservoirs. We are interested here in the irreversibility of trajectories in more general models, and especially those featuring asymmetric interactions, since that is a widespread feature in models of biological systems or in asset pricing models in quantitative finance. In particular, we do not adopt a Hamiltonian description of work and heat or a microscopic reversibility assumption, and the detailed fluctuation theorem (8) is, properly, not a theorem but itself a definition of irreversibility.

We study time series resulting from a discretization with observational time τ of continuous stochastic processes. Importantly the underlying bipartite process appears, at limited time resolution, as a non-bipartite process. As a consequence, there is no general convergence of the time series irreversibility to the physical entropy production except for special cases such as Langevin systems with nonzero constant diffusion coefficients. Our mapping irreversibility (8) is the Markovian approximation of the time series irreversibility definition given in [26]. While it is well defined for any stationary process, it describes the statistical properties of long time series only in the Markovian case.

For general interacting dynamics like (1) we are not able to provide a more significant lower bound to the mapping irreversibility than the standard II Law of thermodynamics (10). A more informative lower bound on the mapping irreversibility is found for signal-response models described by the absence of feedback. This sets the backward transfer entropy as a lower bound to the conditional entropy production, and describes the connection between the irreversibility of time series and the discrete information flow towards past between variables.

Importantly, the relation between irreversibility and information measures is not given in general for time series because the results on continuous bipartite systems do not generalize to the time series irreversibility. It appears exactly because of the absence of feedback, and of the corresponding non-complete causal representation. We restrict ourselves to the bivariate case here, but we conjecture that fluctuation theorems for multidimensional stochastic autonomous dynamics should arise in general as a consequence of missing arrows in the (non-complete, see e.g., Figure 2) causal representation of the dynamics in terms of Bayesian networks.

In our opinion, a general relation connecting the incompleteness of the causal representation of the dynamics and fluctuation theorems is still lacking.

Finally, let us note that exponential averages such as our integral fluctuation theorem (18) are dominated by (exponentially) rare realizations [53], and the corresponding II Law inequalities such as our (19) are often poorly saturated bounds. In the receptor-ligand model discussed in section II.B the backward transfer entropy lower bound is roughly one half of the mapping irreversibility, and this is also the case in the BLRM for large τ where the ratio converges exactly to $\frac{1}{2}$. This limitation is quite general, see for example the information thermodynamic bounds on signaling robustness given in [54].

We also introduced a discussion about the observational time τ in data analysis. In a biological model of receptor-ligand systems we showed that it must be fine-tuned for a robust detection of the irreversibility of the process, which is related to the concept of causation [30] and therefore to the efficiency of biological coupling between signaling and response.

## Figures and Tables

**Figure 1 entropy-21-00177-f001:**
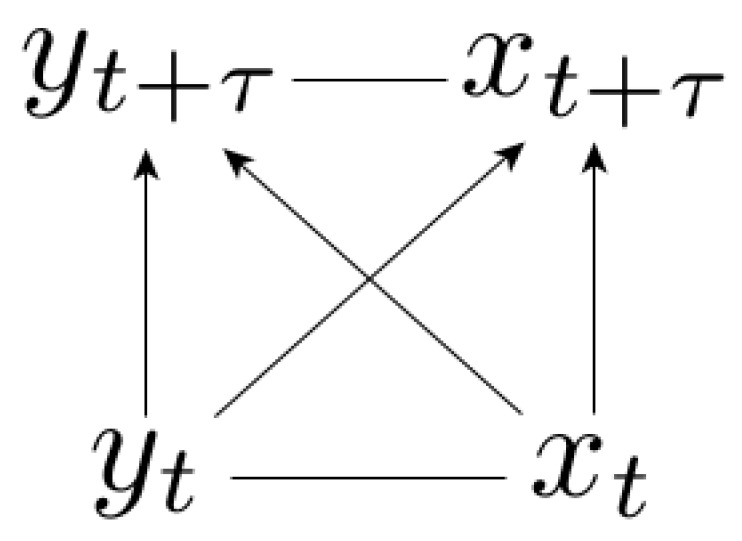
Complete causal representation. The arrows represent the way we factorize the joint probability density. In the complete case the causal representation is fully degenerate: $p\left(\zeta_{\tau }^{xy}\right)=p\left(x_{t},y_{t}\right)\cdot p\left(x_{t+\tau },y_{t+\tau }|x_{t},y_{t}\right)=p\left(x_{t},y_{t}\right)\cdot p\left(x_{t+\tau }|x_{t},y_{t}\right)\cdot p\left(y_{t+\tau }|x_{t},y_{t},x_{t+\tau }\right)=p\left(x_{t},y_{t}\right)\cdot p\left(y_{t+\tau }|x_{t},y_{t}\right)\cdot p\left(x_{t+\tau }|x_{t},y_{t},y_{t+\tau }\right)$.

**Figure 2 entropy-21-00177-f002:**
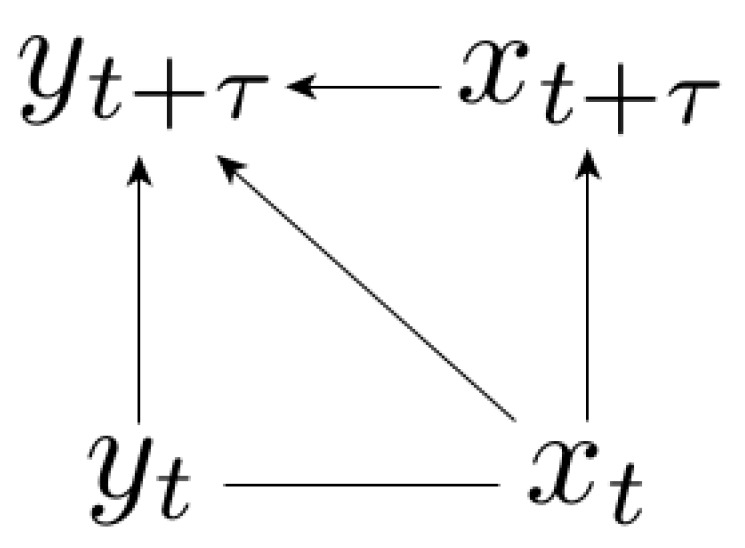
Causal representation of signal-response models. The joint probability density is factorized into $p\left(\zeta_{\tau }^{xy}\right)=p\left(x_{t},y_{t}\right)\cdot p\left(x_{t+\tau }|x_{t}\right)\cdot p\left(y_{t+\tau }|x_{t},y_{t},x_{t+\tau }\right)$.

**Figure 3 entropy-21-00177-f003:**
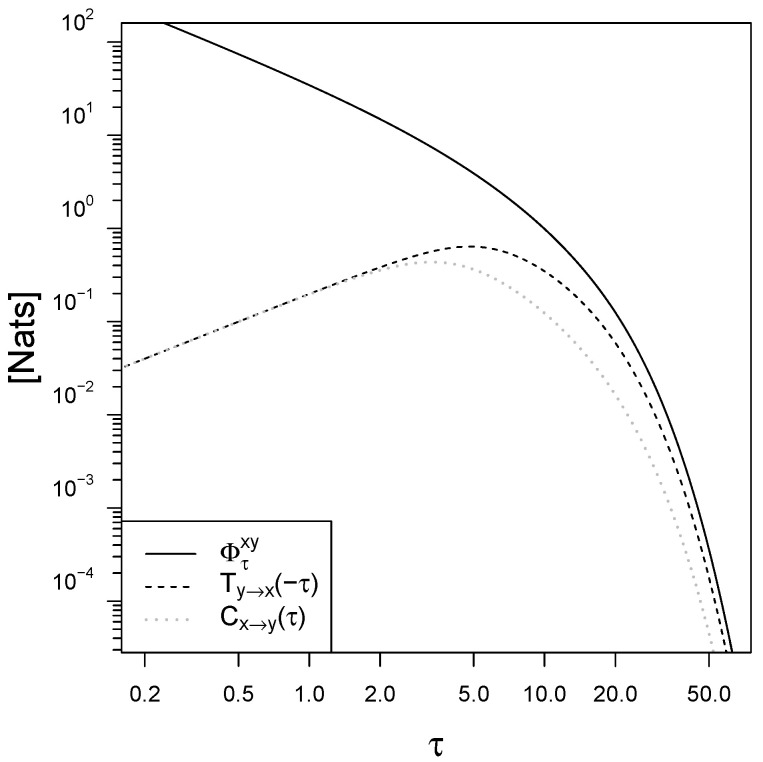
Mapping irreversibility $\Phi_{\tau }^{xy}$, backward transfer entropy $T_{y\to x}\left(-\tau \right)$ and causal influence $C_{x\to y}\left(\tau \right)$ in the BLRM as a function of the observational time interval τ. The parameters are β=0.2 and $t_{rel}=10$. All graphs are produced using R [41].

**Figure 4 entropy-21-00177-f004:**
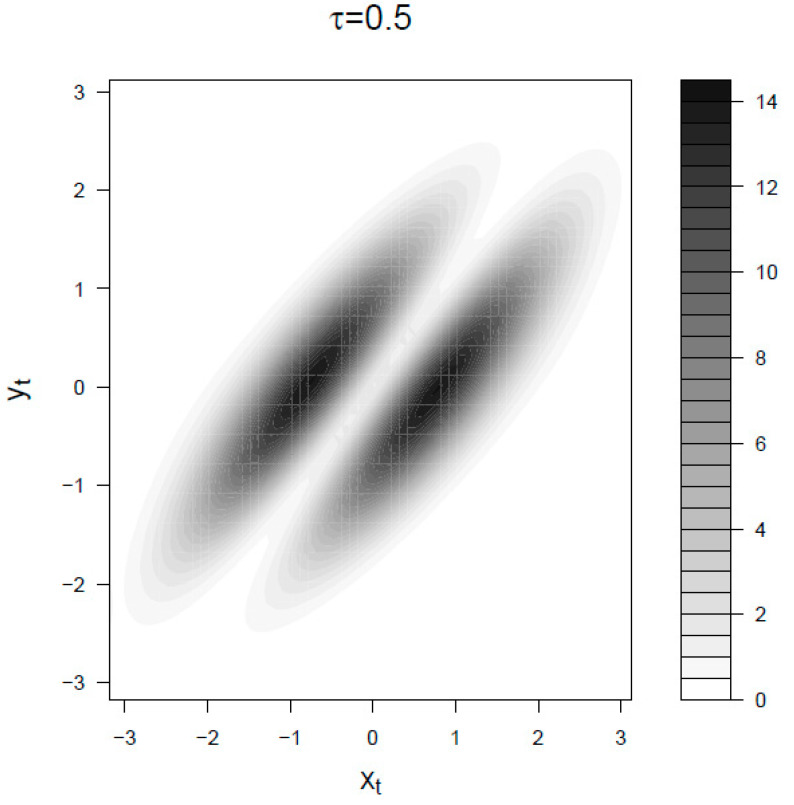
Mapping irreversibility density $\psi (x_{t},y_{t})$ for the BLRM at $\tau =0.5<\frac{1}{\beta }<t_{rel}$. The parameters are β=0.2 and $t_{rel}=10$. Both $\psi (x_{t},y_{t})$ and the space (x,y) are expressed in units of standard deviations.

**Figure 5 entropy-21-00177-f005:**
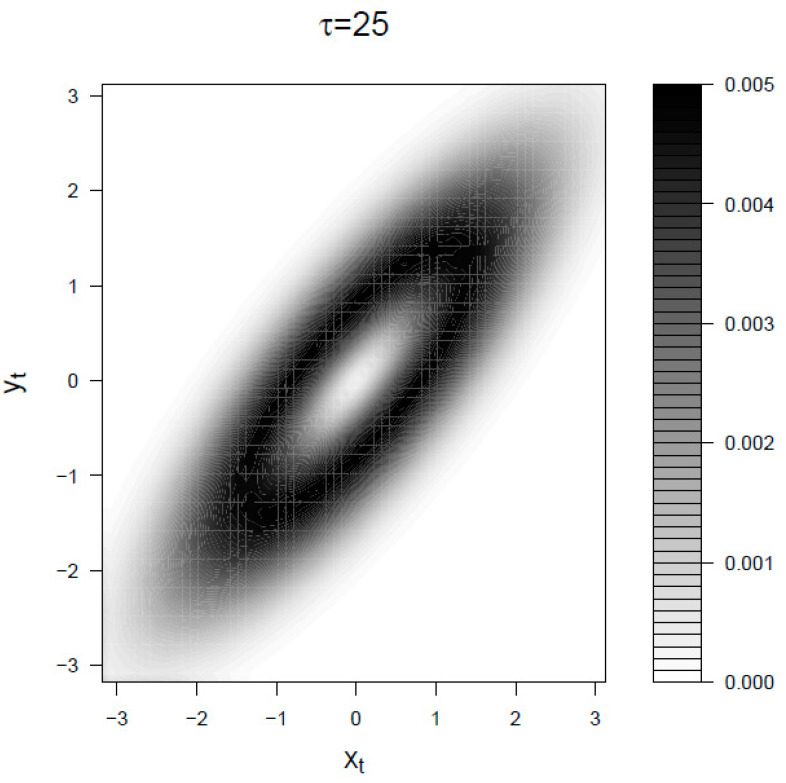
Mapping irreversibility density $\psi (x_{t},y_{t})$ for the BLRM at $\tau =25>t_{rel}>\frac{1}{\beta }$. The parameters are β=0.2 and $t_{rel}=10$. Both $\psi (x_{t},y_{t})$ and the space (x,y) are expressed in units of standard deviations.

**Figure 6 entropy-21-00177-f006:**
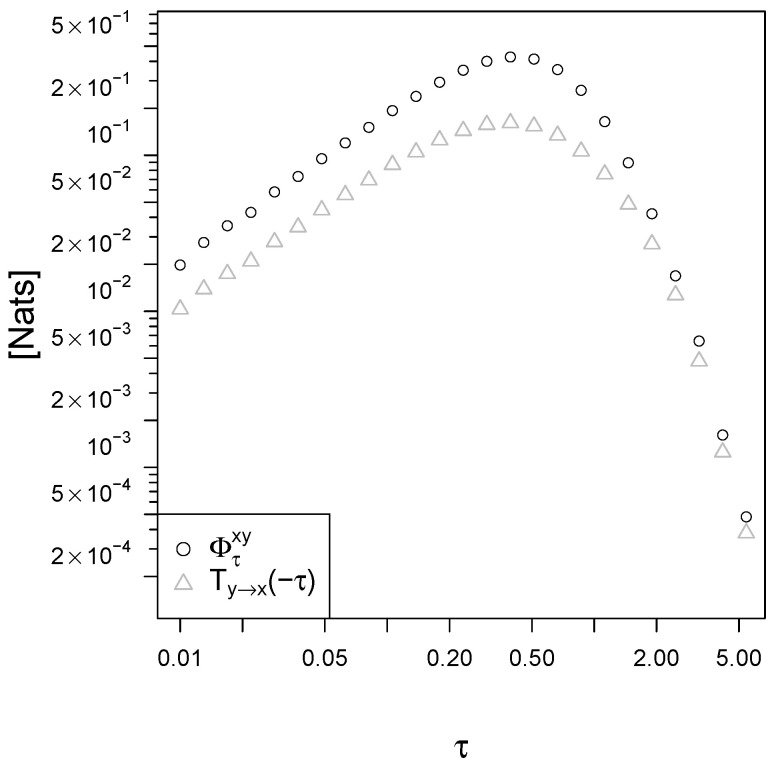
Mapping irreversibility and backward transfer entropy in our model of receptor-ligand systems (Equation (22)). The parameters are $k_{on}=5$, $k_{off}=1$, and h=2.

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
