# Peer review of "Information Thermodynamics for Time Series of Signal-Response Models"

_entropy, 2019, doi:10.3390/e21020177_

Round 1

Reviewer 1 Report

Referee report

A. Auconi, A. Giansanti, and E. Klipp

"Information thermodynamics for time series of signal-response models"

The authors consider systems driven into non-equilibrium steady states in continuous time, but which are observed at discrete points in time ("time series"). Even if the original system is bipartite, the data in such time series appear to be non-bipartite.  The authors generalise information thermodynamic measures for the irreversibility of the system to this setting and derive fluctuation relations for these quantities. Particularly remarkable is a fluctuation relation for signal-response models.  The results are later on illustrated for an analytically solvable model and a more complex, numerically treatable model.

The results presented in the manuscript are interesting and deserve publication in entropy. In order to improve the clarity of the manuscript, I would ask the authors to address the points listed below.

(in chronological order)

1. In the introduction, mathematical notation (phi, T) is used for quantities that has not yet been properly defined. It may be sufficient and less confusing to explain these concepts using words only.

2. 2nd line above (4): the definition of i_t^x and i_t^y is not self-evident

3. l. 103 and below: the concept of extracted work is not directly applicable to the system at hand, which is not formulated in physical terms (e.g. no (free) energy is associated with states x and y). The connection to Ref. [19] should therefore be addressed more informally, in  particular I do not see the necessity to introduce the notation W_ext, which is never used again.

4. above (6) "under the integral sign". It is not obvious which integration is meant here.

5. l. 132 and 137. It might be more correct to use "pair" instead of "couple"

6. Eq. (7) and below: I do not understand the notation for the entropy production. Why is it just Delta and not "Delta s", in analogy to the entropy in the bath. If this is a typo, please make sure to correct it throughout the manuscript. Check also the occurence of the change of bath entropy without "s" in l. 192. 

7. It should be made clear that y_{t+tau} does not really depend causally on x_{t+tau}, as Fig. 1 might suggest. The two are merely correlated, and it is chosen to represent this correlation as if it were a causal one. Since the system is symmetric in x and y, one could equally well have drawn the arrow between the two the other way round. This choice is in my view a crucial logical step and should be explained carefully in the main text.

8. l. 179: typo (probably). Should the index of g and D be x rather than y?

9. Triple equal sign for definition in (11)

10. l. 183: typo "be consider"

11. l. 205 and 209: s_m is not defined. Should this be s_b?

12. Does the fluctuation relation (18) hold only for stationary processes or also for transient ones?

13. l. 254: Check grammar in this sentence. What is the subject for "that are"? What is meant by being "robust to time series"?

14. Paragraph starting in l. 260. The formulation here is very colloquial, and rather imprecise. E.g. "makes the difference": to what exactly?

15. In 5.1., can an interpretation be given for the limit t->infinity? Are the limits of Phi/T and Phi/C universal or specific for the BLRM? It should be noted that the model has only one free parameter (since the units for x,y and t can be chosen freely).

16. The "causal influence" is never discussed in the general part of the manuscript (except for the introduction). Is it necessary here?

17. The values of the parameters alpha and D are not given in the figure captions.

18. Since the ratio Phi/T turned out to be interesting for the BLRM: what is its limit for the receptor-ligand system?

19. Sec. 5.2 ends with "we do not treat this problem here". Why not? It looks like in Fig. 6, Phi and T do converge to a linear behaviour. Would this not allow one to read off the entropy rate? Otherwise I do not see the point in raising this question.

20. l. 364: Grammar: "these does not"

21. l. 524 "moltiplicative"

Author Response

Please find our responses to Reviewer I in the document Modifications.pdf uploaded here.

Reviewer 2 Report

I would like to point out that the content of the manuscript is to a great extent essentially the same as the content of an article of the same authors published on arXiv.org in March 2018: “A fluctuation theorem for time-series of signal-response models with the backward transfer entropy”.

In the manuscript, the authors discuss link between irreversibility and backward transfer entropy. They study a linear signal-response model and a nonlinear biological model of receptor-ligand system to characterize cases in which more informative lower bounds on the  irreversible entropy production can be set.

My comments and questions are as follows.

63 - 64

The „causal influence“ is several times mentioned and it is plotted on Figure 3.  However, it is not defined. It is just mentioned that it was proposed in [30] in the framework of linear Langevin networks without feedback (linear response models). If the causal influence is important for this study, then it would be better to have the exact definition here. But is it necessary here, if the core of the work seems to be based on the backward transfer entropy?

110      It should be “satisfies”.

figure 1 and figure 2

Should not be an arrow from x_t to y_t?

183      It should be “be considered”.

205 and 209

            What is s_m?

379-380 The sentence probably needs to be reformulated due to grammar.

Author Response

Please find our responses to Reviewer II in the document Modifications.pdf, from page 3.

Round 2

Reviewer 1 Report

I agree with all the changes in response to my previous comments, the paper should now be ready for acceptance.

Reviewer 2 Report

I have accepted the author response.